# Stroke and Athletes: A Scoping Review

**DOI:** 10.3390/ijerph181910047

**Published:** 2021-09-24

**Authors:** Patricia K. Doyle-Baker, Timothy Mitchell, K. Alix Hayden

**Affiliations:** 1Human Performance Lab, Faculty of Kinesiology, University of Calgary, Calgary, AB T2N 1N4, Canada; timothy.mitchell@ucalgary.ca; 2School of Architecture, Planning, and Landscape, University of Calgary, Calgary, AB T2N 1N4, Canada; 3Alberta Children’s Hospital Research Institute, University of Calgary, Calgary, AB T3B 6A8, Canada; 4Libraries and Cultural Resources, University of Calgary, Calgary, AB T2N 1N4, Canada; ahayden@ucalgary.ca

**Keywords:** athletes, artery dissection, cerebrovascular accident or stroke, headaches, head trauma

## Abstract

Stroke (i.e., cerebrovascular accident) affects one in 10,000 people between the ages of 14 and 45; however, very little is known about the frequency and type of stroke that occurs in athletes. The risk of injury to the neurovascular structures may depend on the type of sport involvement, although, sport-specific incidence rates are not known. Therefore, the goal of our scoping review was to provide some guidance to better inform the development of a context-fit stroke model by summarizing studies on a broad research topic related to stroke or cerebrovascular accident in sport based on a strict athlete definition. We used the guidance of Arksey and O’Malley’s five-stage-process for a scoping review. Databases included MEDLINE(R) Epub Ahead of Print, In-Process & Other Non- Indexed Citations, Ovid MEDLINE(R) Daily and Ovid MEDLINE(R), and Embase (OVID databases); CINAHL Plus with Full Text, SportDiscus with Full Text (Ebsco databases); and Scopus. Publication dates were from 1979–2020 across nine different countries resulting in 39 individual cases of stroke with an athlete age range of 14–56 years (95% male). The major inciting event(s) prior to stroke onset were headaches (38.4%), head trauma (30.7%), and neck injury and/or vertebral artery dissection (20.5%). Several sporting activities were represented with American football as the most prevalent (30.7%). In summary, we found that sports with an aspect of impact, collision, or microtrauma can lead to subsequent stroke. These sport-related traumatic events were often difficult to diagnose because of the longer interval before ischemia occurred. Therefore, health care providers should be particularly attuned to the possibility of stroke when evaluating athletes presenting with or without neurological deficit.

## 1. Introduction

Stroke (i.e., cerebrovascular accident), occurs in about one in 10,000 people under the age of 64, with hospital admissions currently rising in the age group between 20 and 59 years old [1]. Limited information exists, however, on the frequency and type of stroke that occurs in athletes. McCrory (1999) noted that several articles state that 10% of all stroke patients are missed at first medical contact, and that many of these are seemingly in young healthy adults, accounting for an estimated 3% of strokes [2]. Rohr and colleagues (1996) identified cardiac embolism, hematological stroke, and lacunar stroke as the most common causes in young adults [3]. However, 1/3 strokes suffered by young and fit individuals are not accompanied by symptoms and therefore strokes are under identified.

Sport-specific incidence rates are unknown, perhaps because stroke in sport is a rare clinical entity [4]. The risk of injury to the neurovascular structures may depend on the type of sport involved, and cervicocerebral artery dissection is often identified as the main cause in young athletes [5]. Examples of athletes who suffered a stroke are often found in grey literature such as news reports or blog postings, with the most recent article describing nine athletes under 40 [6]. Unfortunately, many of these examples do not move forward into case reports which could provide descriptive information about the athlete scenario. Case reports are deemed important because they foster an educational platform for describing new and rare diseases, such as stroke in sport. This approach to publishing is considered by many as a valuable resource of unusual information that leads to new research, and management advances in clinical practice [7]. Regardless of these approaches there still exists a limited understanding of the risk factors and injury mechanisms associated with athletes who have a stroke on or off the field of play.

Much of the general prevention research on reducing stroke risk focus’ on a continuous lifetime of physical activity or beginning these activities during later adulthood and, therefore, sport activity solely during young adulthood would be insufficient to reduce risk [8]. Based on this information, our goal was to provide some guidance to better inform the development of a context-fit stroke model, by summarizing the studies on a broad research topic related to stroke or cerebrovascular accident in sport based on a strict athlete definition.

## 2. Materials and Methods

Scoping reviews provide a means of mapping key concepts in an area [9]. Arksey and O’Mally first introduced a five-stage process to guide the conduct of scoping reviews in 2005 [10]. Levac, Colquhoun, and O’Brien advanced the scoping review methodology by providing clarification and enhancements for each of the five stages. They also introduced a sixth stage which suggested consultation with stakeholders for knowledge dissemination [11]. Recently, in 2015, Peters et al., under the umbrella of the Joanna Briggs Institute (JBI), further clarified and standardized the methods associated with conducting a scoping review and our review, then, was informed by Peters et al.’s guidance on conducting scoping reviews [9].

The protocol was developed a priori to guide our scoping review; however, it was not registered as Prospero does not accept scoping reviews. Our protocol is available from the first author. We used the JBI mnemonic PCC: Participants (active athletes), Concept (stroke), Context (sports) to inform our search and data charting. This review is reported in accordance with the Preferred Reporting Items for Systematic Reviews and Meta-Analysis for Scoping Reviews Extensions for Scoping Reviews (PRISMA-ScR) [12].

### 2.1. Definitions

English-language studies focused on active athletes that suffered a stroke were included. For the review, active athletes were defined as any of the following: training in sports to improve performance; actively participating in sports competitions, formally registered in a local, regional, or national sports federation; sports training and competition as a major activity [13]. According to Abbott et al., (2017), the meaning of “stroke” for several decades (in scientific and lay literature) has most often been consistent with the 1980 World Health Organization (WHO) definition [14]. This definition was referenced from an article in 1980 by Aho and colleagues as “rapidly developed clinical signs of focal (or global) disturbance of cerebral function, lasting more than 24 h or leading to death, with no apparent cause other than of vascular origin” [15] (p. 114). Given this, our definition of stroke included at the simplest level when blood flow to the brain is hindered, to temporary (i.e., TIA) or prolonged vascular-related episodes of brain dysfunction, such that the sudden death of some brain cells occur due to lack of oxygen when the blood flow to the brain is impaired by blockage or rupture of an artery to the brain. All study designs were included, as well as editorials, letters, and opinions with no date or athlete age limit imposed. Studies that focused on non-athlete populations, or non-active athletes (e.g., retired) were excluded. Further, studies using sports as a form of rehabilitation were not included. Studies that did not include stroke, as defined above, were also excluded. All reviews including systematic, scoping and narrative were excluded.

### 2.2. Information Sources and Search Strategy

Prior to commencing the research, an exploratory search of the Prospero, Cochrane Database of Systematic Reviews and the JBI EBP Database was conducted to determine if knowledge synthesis studies relevant to our aim and research questions had been previously conducted. We did not find any reviews or protocols.

A comprehensive search strategy was developed in collaboration with the research team, led by the health sciences librarian (KAH). The search focused on two main concepts: stroke and athlete/sport. Both subject headings and keywords were used in each database. A draft search was first developed in Medline based on an analysis of known relevant studies. Given the term “stroke” is often used in sports literature, we mindfully excluded sports-related stroke terms in the search, for example, golf stroke and swimming stroke. We also excluded medical terms such as heat stroke and stroke volume. All searches were limited to the English-language only.

### 2.3. Literature Search

The following five databases were searched from inception until May 2020: OVID databases: MEDLINE(R) Epub Ahead of Print, In-Process & Other Non-Indexed Citations, Ovid MEDLINE(R) Daily and Ovid MEDLINE(R), and Embase; EbscoHost databases: CINAHL Plus with Full Text, SportDiscus with Full Text; and Scopus. The search was first developed in Medline, and then was translated and adapted for each database. Keywords were the same across all databases, and subject headings were determined by the database indexing (See Appendix A for the search strategies for each database). SportDiscus included grey literature, such as popular press and sport-related news and magazines.

### 2.4. Study Selection

Database results were exported and uploaded into Covidence, (Covidence, Melbourne, Australia). Deduplication was conducted through Covidence’s automatic deduplication function. Before study selection, a calibration exercise was conducted with two researchers (TM, PKDB). A total of 500 studies were pilot tested by screening the titles/abstracts and applying the pre-determined inclusion/exclusion criteria. The inclusion/exclusion criteria were clarified when required, and disagreements were discussed. Screening by titles and abstracts was then conducted by a single reviewer (TM). Cochrane Handbook for Systematic Reviews notes that “it is acceptable that this initial screening of titles and abstracts is undertaken by only one person” [16]. Full-text studies were screened by both reviewers (TM, PKDB), and disagreements were resolved by consensus. The full-text screening occurred during the months of May and June 2020.

### 2.5. Charting the Data

The data extraction instrument for source details, characteristics, and results extraction was modeled from the template in the JBI Manual for Evidence Synthesis [17]. The additional description of an inciting event (i.e., proximal to the outcome) was added, as this is the last link in the web of causation related to injury prevention in sport [18]. The reviewers (TM, PKDB) spent a considerable amount of time charting the data in excel from the retrieved case reports and fact-checking several sources from articles in magazines or newspapers to develop key categories based on standard information from each. This was completed so that the outcome would be contextualised and speak to a broad readership; Arksey and O’Malley refer to this as a ‘descriptive-analytical’ method [10]. Critical appraisal is not a requirement of studies included in a scoping review, nor is it recommended [9]. Therefore, we did not critically appraise the included studies.

## 3. Results

### 3.1. Search Results

Our initial search identified 6081 studies, and this was followed by the removal of duplicates (1842). From the 4239 retrieved records (4083 of which did not meet the inclusion criteria), 156 were transitioned to full-text screening, resulting in 36 records with 39 cases meeting the inclusion criteria. A review flow diagram (see Figure 1) details the search results and study selection approach.

### 3.2. Characteristics of Included Cases

Publication dates ranged from 1979–2020 and originated across nine different countries including Australia, Brazil, Canada, China, Israel, Japan, Turkey, the United Kingdom (UK), and the United States of America (USA). The age of athletes ranged from 14–56 years of age, and 95% were male (37/39). Several sporting activities relating to stroke were represented, with American football (12/39, 30.7%) as the most prevalent. This was followed by four cases in running/marathon (10.2%), three cases in each of baseball and wrestling (7.7%), two cases in each of Aussie Rules Football, boxing, ice hockey, rugby, and soccer (5.1%), and one case in each of basketball, Brazilian Jiu-Jitsu, taekwondo, table tennis, tennis, and volleyball (2.6%).

### 3.3. Inciting Events Prior to Stroke

Table 1 (Case Reports and Articles) describes the case report characteristics and lists each by year, author, country, sport, athlete age, and inciting event description. The top three inciting event(s) before the onset of stroke in athletes were self-reported headache with 15 cases (38.4%), the occurrence of head trauma in 12 cases (30.7%), and some form of neck injury and/or vertebral artery dissection in 8 (20.5%) cases. Other notable inciting events included: genetic predisposing conditions (12.8%), thrombo-embolic clots (7.6%), and hydration suspected complications (5.1%). Overlap of inciting events did occur in some cases.

### 3.4. Vascular Mechanisms of Stroke

Not all articles cited [19,20,21,22,23,24,25,26,27,28,29,30,31,32,33,34,35,36,37,38,39,40,41,42,43,44,45,46,47,48,49,50,51,52,53,54] in the Table 1 described the vascular mechanism of stroke [47]. Many strokes, however, were identified as ischaemic in nature (~57%), with 7 cases (17.9%) of vertebral artery dissection, 5 cases (12.8%) of each intracranial artery dissection, cerebral infarction, and subarachnoid hemorrhage, and 11 cases (30.7%) with neck trauma leading to dissection.

## 4. Discussion

This scoping review aimed to contribute and provide guidance to the ongoing development of a context-fit stroke model by summarizing the studies on a broad research topic related to stroke in sport based on a strict athlete definition. Prevention for stroke is known to broadly encompass three areas. These areas include lifestyle or modifiable risk factors, genetic predispositions or nonmodifiable risk factors, and trauma prevention factors such as sport or stroke triggers, which is a relatively new area of investigation according to Boehme et al., (2017) [55]. In this discussion these areas are used to categorize the cases which are further grouped based on study commonalties. This is a followed by a summary of the case results highlighting some of the inciting events prior to stroke in active athletes, more frequent sport triggers, and inaccuracies in reporting of stroke.

### 4.1. Lifestyle Risk Factors

Maron and colleagues stated that the highly conditioned athlete epitomizes the healthiest segment of our society [56]. The endurance athlete, for example, embraces a lifestyle that promotes cardiovascular adaptations resulting in an increased maximal oxygen intake and cardiac output, with little change or a decrease in maximal heart rate [57]. However, athletes can also choose to partake in risky lifestyle choices including the use of performance enhancing drugs as well as recreational drug and alcohol consumption, cigarette smoking or supplements and dietary choices which may increase the risk for cardiovascular events outside sport [58,59,60].

#### 4.1.1. Performance Enhancing Drugs

The National Institute on Drug Abuse (NIDA) states that serious and life-threatening adverse effects from anabolic steroids are likely underreported, although artery damage and stroke have been reported in athletes younger than 30 years [61,62,63,64]. Many steroid users also use other drugs, making it difficult to show that anabolic steroid use alone causes disability and/or death [65]. Demartini and colleagues (2017), reported on a Brazilian athlete competing in Jiu-Jitsu, presenting to the emergency room (ER) with headache, right motor deficit, and aphasia, one week after practicing a submission maneuver known as the Rear Naked Choke or Lion Killer [41]. The athlete was noted to have hypodensity in the left cerebral hemisphere, several thrombotic occlusions throughout the left middle cerebral artery, and internal carotid artery dissection. Although the athlete had never smoked cigarettes, they admitted to the use of anabolic steroids (nandrolone and trenbolone) one month before the event competition. The authors describe the risk of the submission maneuver and the biomechanical sequences associated with this adverse outcome. They do not however, refer to the possibility of the thrombotic burden from anabolic steroid use previously identified in the literature [41,66].

#### 4.1.2. Supplements

Supplements with contents and label claims are not always evaluated by the US Federal Drug Administration (FDA), but they do have a widespread following and use by athletes for enhancing athletic performance [67,68,69]. The most common causes of death with supplements involve ephedrine use and are related to stroke and myocardial infarction [70]. Health Canada cautioned in 2002 that supplements containing ephedrine used to increase energy or promote weight loss predisposed individuals to negative cardiovascular effects such as vasoconstriction, vasospasm, shortening of cardiac refractory periods allowing re-entrant cardiac arrhythmias, hypertension-induced subarachnoid hemorrhage, and sympathomimetic-induced platelet activation [71,72]. A 24-year-old varsity football player who ingested ephedrine containing supplements was presented by Foxford et al., (2003) [29]. The athlete subsequently developed a vasospastic stroke during training and upon hospital admission, a transcranial Doppler ultrasound revealed a vasospasm of the right middle cerebral artery. The authors summarize that “we must educate athletes about hidden ingredients in supplements and the serious health risks associated with their use (and abuse) for performance” [29]. Athletes must be educated on the differences between approved products with a drug identification number (DIN) and supplements that are not evaluated, particularly those containing ephedrine or ephedra given the potential of serious adverse effects [71]. Health care providers must also be able to educate athletes on avoiding the use of these products given the risk outweighs the benefit, so that athletes continue to perform at optimal levels in a safe and healthy manner [73].

#### 4.1.3. Hydration Status

In sport, dehydration status has been a major topic for many years and often has been associated with the cause of collapse in marathons [74]. The homeostasis of sodium is important for the appropriate functioning of the body and disturbances in electrolyte balance have been attributed to comorbidities of stroke [75]. In this older publication by Phillips et al., (1983), they present the case of a 14-year-old healthy male, who participated regularly in sporting activities and trained specifically for a 13 mile run [20]. The runner collapsed 10 miles into the event with right-sided weakness. A subsequent CT scan of the brain revealed several low-density areas in the left middle cerebral artery suggesting ischemia or infarction [20]. The young runner recovered post-treatment and the authors proposed that the neurological complications were probably associated with excessive dehydration and an altered packed cell volume. No other history was reported, or cause identified, despite the temporary neurological complication [20]. In another case, a marathoner in his 30′s collapsed following rehydration, went into respiratory arrest, became hyponatraemic, and died of hydrocephalus and brain stem herniation [31]. In a follow-up history, it was flagged that the runner had experienced a severe headache three months earlier following an increase in his running schedule. Despite symptoms, he completed a 22 mile run two months later and experienced another period of severe headaches and nausea which was attributed to dehydration. One month later he finished a marathon, collapsed, and died. The authors state that the brain herniation was a likely consequence of hyponatraemic brain swelling on a background of hydrocephalus secondary to a previous subarachnoid hemorrhage (SAH). In both cases, hydration status is the unlikely cause of stroke, even though dehydration may be considered an independent predictor of outcome after ischemic stroke [76]. Tucker (2013) proposes that dehydration has become the ever-present scapegoat in everything in sport [77] and therefore it is important to review the collective evidence from case studies [78]. A prior history should not be overlooked and should be considered in combination with early identification of altered serum sodium and acute hypohydration, both possible mechanisms that accompany stroke in endurance athletes [79,80].

### 4.2. Genetic Risk Factor

Genetic predispositions can put individuals at a much greater risk for stroke and therefore should be documented for improved identification and education [81]. Several cases describing undiagnosed or undiscovered genetic abnormalities which subsequently led towards a stroke were identified [38,45,46,47,52] and many were only reported in news-type literature such as newspapers [54]. Three separate cases of high-level athletes with undiagnosed holes in the heart (i.e., patent foramen ovalem [33,54]; atrial septal defect, [53] contributed to adverse cardiovascular events). For example, Corhern and colleagues (2017) describe the case of a junior linebacker, who was discovered to have a patent foramen ovale following a stroke [54]. The linebacker thought it was “just a headache” and he could “sleep it off”, although the accompanying numbness, balance issues, and tingling prompted the athlete to reconsider. Extreme physical effort can be a trigger for serious, and often fatal, cardiovascular events in athletes with previously undetected underlying heart or vascular disease. Amateur athletes, as in this case, of college football, typically undergo a fitness test [54]. The fitness test, however, may not be an extreme physical effort for a well-conditioned athlete and therefore not stressful or comprehensive enough to detect an underlying heart or vascular genetic condition. The prevalence of patent foramen ovale is quite high in the population (25% or one in four) [82], and therefore it is difficult to identify if, in this case example, it was the cause of stroke or was incidental.

#### Genetic and Lifestyle Risk Factor

Genetic background paired with lifestyle risk factors in a sport where weight (i.e., a high Body Mass Index (BMI)) is valued increases the opportunity for an adverse event. A 28-year-old American football lineman who experienced an ischaemic stroke following a game is an example of this combination [24]. The CT revealed a small hypodensity in the left centrum semiovalve (internal capsule) suggestive of ischemic etiology. An axial T2 weighted MRI followed which revealed hyper intensity and swelling of the left basal ganglia consistent with an acute infarct (Lacunar Infarct). The athlete weighed approximately 300 lbs, was on medication for hypertension since the age of 18, and both parents were hypertensive. This type of stroke is considered unusual at this age and therefore the contributory roles of hypertension, high BMI, and extreme exertion from playing professional football must be considered as contributing factors [24].

### 4.3. Sport and Stroke Triggers

#### 4.3.1. Rugby Football

Schneider once stated, (1964) “There is probably no better experimental or research laboratory for human trauma in the world than the football fields of our nation” [83]. Specific sporting environments such as football can predispose athletes to a cardiovascular event or act as a stroke trigger [55,84]. Arterial dissection is the predominant pathophysiological mechanism leading to stroke in the athletic domain according to McCrory (1999) [2]. Case examples of this are presented by McCrory, (2000), where three players from Australian rules football and Rugby league are highlighted because of the in-game head impacts during tackling [28]. Two of the players died shortly after arriving at the hospital, and one athlete presented to the ER with ataxia and slurred speech and was sent home due to a “lack of neurological abnormalities”. This athlete continued to suffer from ongoing symptoms post-discharge and subsequently died in the intensive care unit (ICU) following a return trip to the ER [28]. The failure to adequately detect and manage vertebral artery dissection clearly can lead to neurologic sequelae and/ or death [85,86]. Yet, the variety of clinical presentations make it difficult for physicians to make a correct diagnosis, particularly when confronted by an unconscious athlete, on or off the field of play with the potential of spinal injury, or in the ER with limited symptomatology.

#### 4.3.2. Boxing

Boxing is another sport known to have sequelae of traumatic brain injury (TBI), because of repeated blows directly to the head [87,88] which can produce the same force as being hit with a 6-kg wooden mallet at a striking speed of 20 mph [89]. These continuous blows to the head produce changes in the cerebellar region which can predispose an individual to an acute bleed followed by scarring, atrophy, fine fibrous gliosis, and demyelination with Purkinje and granule cell loss [90]. Kariyanna et al., (2019) recently presented a case of a 26-year-old-male boxer who was knocked unconscious during a match and, although wearing protective headgear and gloves, developed hemorrhaging in the bilateral cerebral hemispheres and was subsequently declared “brain dead” [44]. In this sport, the rule changes in the early 2000s that were devised to promote safety by limiting the number of rounds, mandatory use of safety head guards, and better medical control during the match, seem to be of limited value in preventing head injury leading to cerebral [44] or cardiovascular event [91].

#### 4.3.3. American Football

Athletes who sustain a minor head injury are four times more likely to undergo a second minor head injury, and recently it has been recommended that any player with a head injury be removed from play to allow adequate observation and treatment [25,27]. Recurring minor head injuries have the potential to lead to repeated concussions, referred to as “second-impact syndrome”, rendering the brain susceptible to vascular congestion without edema or pathological lesions [25,92]. Therefore, returning an athlete to play following several concussions, even when not “reporting” persistent symptoms, may bring potential health risks. Recent management has considered prioritizing physical and cognitive rest for a predetermined duration of time post resolution of symptoms [93]. In this scoping review, we found two very similar case reports [23,50] in young football players, both with post-concussion symptoms. Litt (1995) describes a case of a 16-year-old football player with a persistent headache following an in-game collision [23]. A concussion diagnosis was given after the negative CT scan and provocative testing failed to recreate the symptoms. The athlete was cleared for play 30 days later as they denied any symptoms. In a subsequent game the athlete was blindsided and, on the sidelines, initially denied any neurological symptom, however shortly after complained of dizziness, projectile vomited, and became unresponsive. The neurosurgeons diagnosed a right subdural hematoma by CT scan and the post-operative report noted a second area of chronic membrane formation consistent with past head trauma. Kumar and colleagues (2018) suggest CT scanning lacks sensitivity for acute/hyper ischemic stroke, and that a thorough neurological examination alongside close monitoring is fundamental regardless of seemingly normal neuroimaging at the time of presentation [43]. In a post-surgical follow-up interview the athlete admitted having constant headaches and symptoms between the first and second injuries [43].

Athletes not reporting symptoms is a reoccurring theme in the literature that has been particularly noted in high-risk contact sports such as football where an image of toughness has been part of the culture. Recently, education and awareness of concussion etiology and symptoms or initial lack of it has been improving in the younger age groups [94]. Education, however, on headache symptomology, such as a precipitating headache or migraine associated with stroke, may be lacking in athletes based on this review.

#### 4.3.4. Baseball

Athletes involved in a repetitive exercise of the arms, risk developing thoracic outlet syndrome (TOS) since the subclavian artery may be subjected to trauma resulting in intramural hematoma, dissection, or thrombosis of the artery [21]. Baseball players comprise a population that not only has a higher predisposition of the development of TOS, but also have high-performance demands which can hinder recovery following injury [95]. A case highlighting this was reported by Fields and colleagues (1986) and pertained to a 30-year-old right-handed male major league baseball pitcher who subsequently developed stenosis of the right subclavian artery [21]. The player subsequently suffered an embolic stroke and with noted progressive fatigue and heaviness in his pitching arm during the regular season. A similar case of a professional baseball player was published earlier [48] and suffered a stroke following the development of TOS. In this case report the athlete presented with a “dead arm” which is often attributed to internal impingement and superior labral anterior-posterior lesions [96].

The mechanism of cerebral embolization associated with the TOS is both poorly understood, and difficult to identify in the initial stages of the development. According to Meumann (2013), it may be due to retrograde propagation of thrombus or transient retrograde flow within the subclavian artery exacerbated by arm abduction [97]. Awareness of a change in symptom reporting by the athlete is an important consideration for future research. As highlighted by Fields and colleagues, (1986) a progressively shorter time interval from the beginning of daily workout to the onset of symptoms “is an ominous sign of impending disaster” and “therefore investigation and treatment become matters of great urgency” [21].

## 5. Summary

We found that sporting types with an aspect of impact, collision, or microtrauma can lead to subsequent stroke. Most athletes were males, and this curious lack of females may be due to the large number of sports that have limited female representation when compared to males [98]. We had a large age range with athletes as young as 14 years presenting with stroke. Several case reports such as the backyard wrestler [99] and other review studies with no obvious trauma with young athletes [100] were excluded because the authors did not identify training to improve sport performance, part of our rigorous athlete inclusion criteria. Many of these case reports had vertebral arterial dissection, a known cause of stroke in young adults [101], but others did not have conventional vascular risk factors [38]. We suggest, as others do, that currently there is a lack of evidence-based management guidelines for these uncommon aetiologies [102].

The events prior to stroke in these case reports were varied, however a few common trends were found. In a review by Jones et al., (2010) the most identified symptoms of stroke did not include headaches, however, in this review precipitating headaches, i.e., some degree of headache that occurred before the stroke, was reported in 38% of case reports [103]. This was followed by head trauma (30%) and hyperextension or hyper-rotation of the neck (22%), leading to vertebral artery dissection. These cases with unusual neck movement underscore the importance of considering a differential diagnosis in athletes [84,99], since the failure of detection and management of vertebral artery dissection has the potential for neurologic sequelae or death, [83,86]. A recent systematic review echoes these findings [104].

### 5.1. Implications or Future Directions

Although within the screening process for article selection, the term stroke was broadly used, often it was not mentioned, and only technical terminology was used (vertebral artery dissection [105]; cerebral embolization [106]). Medical professionals have been hesitant or reluctant to use the term stroke when both diagnosing and documenting stroke in athletes and we suggest that a consensus should be met on a preferred term for the condition which can be used more universally. Often concussion was the initial diagnosis and ruling out the possibility of more fatal forms of cardiovascular events was initially not considered. Previous studies have suggested that poor recognition of the warning signs of stroke is the main cause of delay in presenting to the ER [103].

### 5.2. Limitations

Our literature search focused on individuals meeting a strict athletic status. Some studies included both older and younger athletes who were not reported as well as those of a lessened athletic status. Examples of this include two amateur boxers who were described as a printer and solider [107], an amateur rugby player described as a taxi driver and smoker [108], a 16-year-old participating in a baseball summer league [109], and three cases of cervical carotid dissection associated with CrossFit workouts [110]. Although we adhered to our strict athlete screening criterion, inherent biases including publication bias may have influenced our results. Only two cases reported on women athletes presenting with a stroke. This may not be surprising given that women are generally older when they have a stroke and therefore their athletic career would have ended several years previously. However, the underlying prevalence of stroke in female athletes may be underreported. Some cases which were older than two decades were difficult to interpret with respect to identifying the inciting event leading to stroke. Therefore, to avoid assumptions we describe the lead up to point in the direction of the inciting event. We are aware that TIA, particularly postural or situational related to sport, could result in an incorrect diagnosis of syncope. In general, individuals who present with bouts of fainting tend to be elderly males with a high incidence of ischemic heart disease and hypertension [111]. Further, we included English language studies only, and recognize that we could have potentially missed relevant studies published in other languages.

### 5.3. Future Research

Genetic factors were quite low in this scoping review compared to symptoms such as headache and/or sport induced trauma, yet often a history was not taken or mentioned. Screening prior to play is a much-debated topic, particularly on whether the inclusion of the ECG and/or cardiac imaging in routine preparticipation sports evaluation is appropriate or warranted [81]. At the very least, a stroke registry dedicated to athletes may help with determining prevalence [112]. Future research should also consider developing a conceptual framework to address common misconceptions related to stroke symptom reporting behaviour in athletes. Similar background work has been completed in concussive symptom reporting behaviours with enhanced clinical and research practices as outcomes [113].

## 6. Conclusions

This scoping review not only highlights that stroke is occurring in athletes but identifies those sports such as American football have a higher incidence. As previously noted by Litt (1995), many head injuries with mild symptomology in “contact sports are ticking time bombs” and serve as a potential warning sign for subsequent declining mental functions that can lead to stroke and death [23]. The influence of these sport-related traumatic events is often difficult to diagnose because of the long interval before ischemia occurs, and as such are labeled as a mild form of a concussion. Future education with a focus on recognizable symptoms and a description of common inciting events prior to a stroke could play a major role in changing attitudes toward stroke prevention in sport. Continuing to advocate for rule changes by ‘raising the index of suspicion’, which includes the biomechanical sequences in sports that lead to arterial dissection, is equally as critical from a future prevention perspective in healthy young athletes [99,114]. As well including a complete medical history accompanying a more liberalized medical screening approach for blunt carotid and vertebral artery injuries as previously identified by Kerwin and colleagues (2001) [115]. Finally, according to Newman-Toker and colleagues (2014), strokes are misdiagnosed disproportionately in those presenting with headache or dizziness [116]. In this review, headache was the most reported symptom, therefore health care providers should be particularly attuned to the possibility of stroke when evaluating young athletes presenting with this symptom, with or without neurological deficit.

## Figures and Tables

**Figure 1 ijerph-18-10047-f001:**
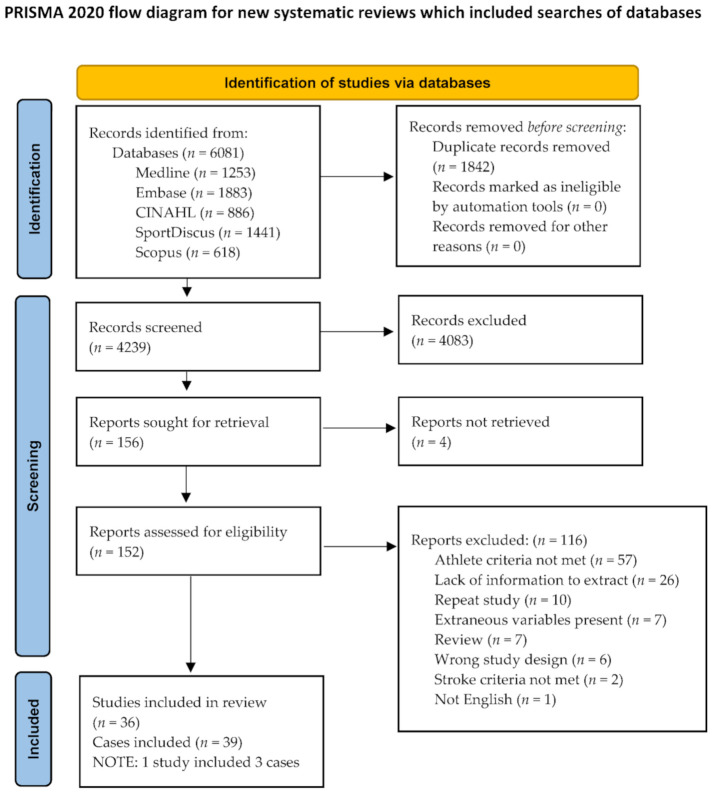
Screening of records on stroke in athlete. Page, M.J.; McKenzie, J.E.; Bossuyt, P.M.; Boutron, I.; Hoffmann, T.C.; Mulrow, C.D.; et al. The PRISMA 2020 statement: An updated guideline for reporting systematic reviews. *BMJ* **2021**, *372*, n71, doi:10.1136/bmj.n71. For more information, visit: http://www.prisma-statement.org/ (accessed on 11 September 2021).

**Table 1 ijerph-18-10047-t001:** (**A**) Case Reports (80.5%, 29/36) by year, author, country, sport, age (years), and inciting event. (**B**) Articles (19.4%, 7/36) by year, author, country, sport, age (year), and inciting event.

Year	Author	Country	Sport	Age	Inciting Event
(**A**)					
1979 [19]	Rogers	USA	Wrestling	17	10 days prior to admission participated in neck “bridging” exercises by using the top of the head as a fulcrum. During a wrestling match was placed in several neck holds, and developed vertigo, ataxia, and numbness associated tingling on the left face and body half. Athlete was unable to swallow fluids.
1983 [20]	Phillips	UK	Marathon	14	After 10 miles of running felt unwell, developed heaviness in right leg, and collapsed with right-sided weakness. On hospital arrival was alert although dysphasic with right hemiplegia.
1986 [21]	Fields	USA	Baseball	30	Thoracic outlet syndrome with Raynaud phenomenon: compressed subclavian artery leading to thrombosed/foci
1991 [22]	Weinstein	USA	Football	29	Forced cervical flexion injury; 10 days prior to evaluation.
1995 [23]	Litt	USA	Football	16	Headache following a collision on the last play of the game; continued to complain of a headache the following week. Cleared for play, blindsided in another game followed by dizziness and headache with projectile vomiting; become pale and unresponsive.
1997 [24]	Mann	USA	Football	28	Slurring of speech and tendency to veer to the right post 60 min after a game. Over the next few days slurred speech persisted and experienced a mild headache and lethargy.
1998 [25]	Kersey	USA	Football	19	Mild concussion sustained during a game. Disclosed to trainer had mild to moderate headaches -left sided and sometimes associated with nausea. Persisted for 12 days and 5 weeks later sustained a head injury in game followed by immediate head pain, vertigo, and bilateral leg paresthesia.
2000 [26]	Malek	USA	Kick Boxer	42	Ruptured intracranial dissecting vertebral aneurysm
2000 [27]	Bruzzone	Italy	Soccer	19	Ball headed and crashed, striking right frontoparietal region against the opponent’s left frontal scalp. Fell to the ground, unconscious for 20 to 30 s. 20 min after was alert and conscious without neurologic deficit. Developed amnesia followed by nausea, vomiting, and headache.
2000 [28]	McCrory	Australia	1–2: AFB 3: Rugby League	15; 20; 27	1. Struck on the right mastoid process by an elbow during the collision, collapsed and unconscious, died (subdural and subarachnoid hemorrhage). 2. Left side head blow; unconscious, died (subarachnoid hemorrhage). 3. Neck strike during a tackle, left the field complaining of arm “pins and needles” with deep neck pain. Originally refused medical treatment; became ataxic with slurred speech following morning. Discharged from the Emerg, no neurological abnormalities were found. Sought chiropractic treatment and after 3rd visit suddenly deteriorated with tinnitus, face alteration, slurred speech. Died in Emerg (vertebral arterial dissection)
2003 [29]	Foxford	USA	Football	24	Ingestion of Xenadrine prior to football training. Severe right sided headache and vasospasm.
2004 [30]	DeGiorgio	Italy	Basketball	23	Stiff neck complaint at the beginning of a practice, fell and lost consciousness. Taken to the Emerg. One-week general condition worsened and died.
2007 [31]	Petzold	UK	Marathon	30′s	Athlete suffered from a thunderclap headache, subsequently developed morning headaches. Collapsed following rehydration, went into respiratory arrest, became hyponatraemic and died of hydrocephalus and brain stem herniation.
2009 [32]	Kanwar	USA	Football	17	Sudden onset of slurred speech and drooling during game warm up. Neurological defects were noted on the left side of face along with left-sided extremity weakness. History identified prior injury to the right shoulder 3 months earlier.
2010 [33]	Miyazawa	USA	Volleyball	22	Presented with acute right-side facial droop, aphasia, loss of muscular control and sensation in the right upper extremity during a match.
2010 [34]	Cohen	Israel	Taekwondo	23	During a championship match an attacker‘s heel kick landed on the unprotected nuchal region resulting in severe head trauma: athlete collapsed developed acute respiratory arrest.
2012 [35]	Smith	Unknown	Running	?	50 plus miles per week. On an aborted training run experienced neurological confusion, diagnosed with right internal carotid artery dissection at Emerg Rehab followed with two pulmonary emboli. Some discussion about disrupted sleep architecture.
2012 [36]	Hart	USA	Boxing	23	Headaches from sparring during training; no protective head gear.
2015 [37]	Matsumoto	Japan	Football	23	Helmet collision during game. Experienced mild nausea/dizziness, resolved in 15-min and team director allowed him to RTS the next day. 3 days prior another head collision was encountered.
2016 [38]	Nelson	USA	Tennis	17	Complained of back and left leg pain, followed by flaccid left leg paralysis after 1 week of heavy training. Spinal cord infarct: likely fibrocartilaginous embolism (monoplegia)
2017 [39]	Degen	USA	Ice Hockey	32	Experienced discomfort and dizziness from falling to the ice. Symptoms resolved and continued to play. Two days post, complained of severe headache (frontotemporal) which persisted overnight.
2017 [40]	Esianor	USA	Football	16	Transient loss of consciousness after two simultaneous head-to-head collisions during a game.
2017 [41]	Demartini	Brazil	Brazilian Jiu-Jitsu	27	Presented to the Emerg with headache, right motor deficit, and aphasia, all commencing 16 h earlier. History showed severe neck pain one week earlier while practicing a submission maneuver known as the Rear Naked Choke or Lion Killer, with persistent pain locally thereafter.
2018 [42]	Ellis	Canada	Soccer	16	Attempted to head the ball, fell backwards; head struck the goalpost. Knocked out followed by a 30 s seizure.
2018 [43]	Kumar	USA	Wrestling	16	Headlock and “face-planted” into the mat; dazed and slow to stand. Complained of headache and nausea on route to hospital. Followed by persistent headache and uncontrolled vomiting.
2019 [44]	Kariyanna	USA	Boxing	26	Knocked out: protective head gear slipped; Multiple strikes sustained to face and head.
2019 [45]	Cheng	China	Table Tennis	56	Developed sudden syncope and right hemiplegia within 2 h of a match; admitted to Emerg.
2020 [46]	David	Turkey	Ice Hockey	17	Severe Headache during final minute of game.
2020 [47]	Muthalagappan	UK	Football	30	Seizure-like activity with hand and lip tingling in athlete with a history of Graves’ disease and previous thyroidectomy. Calcitriol, ergocalciferol and high dose levothyroxine were prescribed.
(**B**)					
1980 [48]	Nack	USA	Pro Baseball	30	Dead arm followed by stroke 6 weeks later.
2002 [49]	Cannella	USA	Pro Baseball		Pain in head resulting from cerebral hemorrhage.
2008 [50]	Schmidt	USA	Football	16	3 weeks post-concussion, cleared to play, game tackle resulted in from cerebral hemorrhage.
2010 [51]	Holland	USA	Rugby	27	Series of strokes likely triggered by a clot in vertebral artery which was pinched during a twist to pass the ball to the wing.
2013 [52]	Beresini	USA	Former Football Collegiate/Marathoner	32	Collapsed in a triathlon. Bicuspid aortic valve: congenital disorder.
2016b [53]	Pilcher	USA	Wrestling	20	During a practice developed partial paralysis to left side: facial droop, migraines, and slurred speech.
2017 [54]	Corhern	USA	Football Collegiate	?	Congenital disorder.

(**A**) Note: Aussie Rules Fottball, AFB; Smith (2010)—Chapter—Case Report; Miyazawa (2010) and Nelson (2016) reported on female athletes. (**B**) Note: All males.

## Data Availability

The search is available as a Appendix A.

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
