# Peer review of "Stroke and Athletes: A Scoping Review"

_ijerph, 2021, doi:10.3390/ijerph181910047_

Round 1

Reviewer 1 Report

Dear authors,

First of all, congratulations on such an interesting topic and research.

I have some comments and suggestions in order to help you increase your manuscript's quality and scientific soundness.

ABSTRACT

  1.  The goal described in the abstract isn't the same that's expressed in the Introduction. You should maintain always the same goal throughout the sections.

INTRODUCTION

  1. In this section, you describe your goal as to «document existing evidence (...) to better inform the development». This is Evidence-Based Practice (EBP). If this is your aim (which is not supported by the abstract or induced by your title), you should include some paragraphs talking about current EBP in this topic, which are the main gaps and which ones you're addressing.
  2. It seems somehow unsuitable to specifically mention a person (in this case, Dr. Richard Swartz) as a reference in such a broad topic, without being here to belittle the person in question. It would be acceptable if it was a globally recognized authority or institution commonly cited. To start the Introduction, and mention epidemiologic data as you did, important institutions like World Health Organization (International), American Stroke Association (America), European Stroke Association (Europe) or Heart and Stroke Association of Canada should be cited.
  3. Line 31: typo 'stoke neurologist'
  4. If authors followed PRISMA-ScR to guide the research, the Introduction should include 'an explicit statement of the questions and objectives being addressed with reference to their key elements (e.g., Participants, Concepts, Context)'. The question/objectives mentioned, either in the abstract or Introduction, are too general to be addressed and to perform a scoping review. Your questions/objectives should be in accordance with the Results you've presented, that, in turn, should be aligned with the Conclusion. In this sense, would you think that the following would be suitable for you: 1) Which are the inciting events prior to stroke in active athletes?; 2) Which are the sports that most frequently trigger stroke events?. A third question would follow - 3) What is the prevalence of stroke events in different sports? - buy, although you've mentioned 'frequency' in the abstract, as being a gap in the literature, you didn't solve this in your results. Either disregard this or further explore it.

MATERIALS AND METHODS

  1. Authors should mention if, during the preliminary search in databases, they've found any existing or undergoing revision on the topic.
  2. It seems that the scoping review was based on four different methods: Arksey and O'Malley's (2005); Levac, Colquhoun, and O'Brien's (2010); JBI (Peters et al., 2015); and Cochrane. It's not mandatory to follow just one method, although maybe it would be simpler, but it's not clear how each method contributed to the review, except for Arksey and O'Malley's. Authors should clearly explain the contribution of each method to their research.
  3. The protocol mentioned in line 70, is it published or registered somewhere (e.g., OSF)?
  4. Inclusion criteria are well described. But there are no exclusion criteria.
  5. Please, avoid common expressions as 'aka' or 'etc', as they have no scientific meaning or soundness and shouldn't be used.
  6. On '2.1 Definitions', authors state that they consider the definition of the stroke to include 'the simplest level when blood flow to the brain is hindered'. What about 'syncopes'? As you know, TIA can be incorrectly diagnosed as syncopes, of which the main clinical feature is precisely the decrease in systemic arterial pressure and in the global cerebral blood flow. This is a huge source of bias.

RESULTS

  1. Authors should explain the reason to exclude 4083 reports from their review.
  2. What do you mean by '39 cases meeting the inclusion criteria', if only 36 were included in the review?
  3. The description of the inciting event in Table 2 is difficult to read, as lines are too close to each other. Would it be possible to draw or format the Table for a better reading?
  4. The 7th report (Kersey, 1998): I suggest writing 'Persisted for 12 days', instead of 'persisted for 12 days' and avoid starting sentences with numbers ('5 weeks later').
  5. The 8th report (Malek, 2000), the original authors report the kick boxer to have 42 years. Maybe you could include that.

DISCUSSION

  1. In the first paragraph the authors write an objective that's in accordance with the one presented in the Introduction, but not with the abstract.
  2. Line 255: a closing parenthesis is missing after 'cardiovascular events'.
  3. Line 301: a typo in 'cerebellar region with can predispose'.

SUMMARY

  1. Line 381: a typo in 'hyper-rotation of the neck the (22%), leading'.

CONCLUSION

  1. This section should be revised, taking into consideration the possible changes suggested before, namely in the objectives. 
  2. There should be something related to solutions/answers to the type of stroke and relation to sports.
  3. If EBP is to be maintained, some conclusions on this should also be written.

Author Response

Thank you for your thorough review and suggestions to increase the readability of this manuscript, we recognize and appreciate the time a review takes.

ABSTRACT

  1.  The goal described in the abstract isn't the same that's expressed in the Introduction. You should maintain always the same goal throughout the sections.

Thank you for identifying inconsistencies between the abstract (this scoping review aimed to summarize studies on a broad research topic related to stroke or cerebrovascular accident in sport based on a strict athlete definition) and the introduction. We have changed this in the abstract and introduction (line 56-60) and note this is more in line with the discussion /summary flow as well “Based on this information our goal was to provide some guidance to better inform the development of a context-fit stroke model by summarizing the studies on a broad research topic related to stroke or cerebrovascular accident in sport based on a strict athlete definition."

INTRODUCTION

  1. In this section, you describe your goal as to «document existing evidence (...) to better inform the development». This is Evidence-Based Practice (EBP). If this is your aim (which is not supported by the abstract or induced by your title), you should include some paragraphs talking about current EBP in this topic, which are the main gaps and which ones you're addressing.

We appreciate your perspective on evidence-based practice and potentially we were too ambitious in our wording. We changed this to read “Our goal was to provide some guidance to better inform the development of a context-fit stroke model of care and prevention in sport.”

  1. It seems somehow unsuitable to specifically mention a person (in this case, Dr. Richard Swartz) as a reference in such a broad topic, without being here to belittle the person in question. It would be acceptable if it was a globally recognized authority or institution commonly cited. To start the Introduction, and mention epidemiologic data as you did, important institutions like World Health Organization (International), American Stroke Association (America), European Stroke Association (Europe) or Heart and Stroke Association of Canada should be cited.

It was not our intention to bring negative attention to Dr. Swartz and therefore we agree with your comment and have removed his name and cite the Heart and Stroke Foundation of Canada report on Different Strokes.

“Stroke also called cerebrovascular accident (CVA), occurs in about one in 10,000 people under the age of 64 and hospital admissions are rising in the age group between 20 and 59 years old.”

  1. Line 31: typo 'stoke neurologist'

Noted and deleted.

  1. If authors followed PRISMA-ScR to guide the research, the Introduction should include 'an explicit statement of the questions and objectives being addressed with reference to their key elements (e.g., Participants, Concepts, Context)'. The question/objectives mentioned, either in the abstract or Introduction, are too general to be addressed and to perform a scoping review. Your questions/objectives should be in accordance with the Results you've presented, that, in turn, should be aligned with the Conclusion. In this sense, would you think that the following would be suitable for you: 1) Which are the inciting events prior to stroke in active athletes?; 2) Which are the sports that most frequently trigger stroke events?. A third question would follow - 3) What is the prevalence of stroke events in different sports? - buy, although you've mentioned 'frequency' in the abstract, as being a gap in the literature, you didn't solve this in your results. Either disregard this or further explore it.

Please see the changes associated with your comments-thank you.  “PRISMA-ScR is a reporting guideline, not conducting. We realize we misstated this in the manuscript and have changed the sentence in our manuscript: “This review is reported in accordance with the Preferred Reporting Items for Systematic Reviews and Meta-Analysis for Scoping Reviews Extensions for Scoping Reviews (PRISMA-ScR) [12].”

MATERIALS AND METHODS

  1. Authors should mention if, during the preliminary search in databases, they've found any existing or undergoing revision on the topic.

Thank you we have added  “Prior to commencing the research, an exploratory search of Prospero, Cochrane Database of Systematic Reviews and JBI EBP Database was conducted to determine if knowledge synthesis studies relevant to our aim and research questions had been previously conducted. We did not find any reviews or protocols.“

  1. It seems that the scoping review was based on four different methods: Arksey and O'Malley's (2005); Levac, Colquhoun, and O'Brien's (2010); JBI (Peters et al., 2015); and Cochrane. It's not mandatory to follow just one method, although maybe it would be simpler, but it's not clear how each method contributed to the review, except for Arksey and O'Malley's. Authors should clearly explain the contribution of each method to their research.

Thank you for pointing out the confusion with respect to the various approaches to conducting a scoping review. Arksey and O’Malley’s work was adapted, refined, and modified by Levac et al and subsequently Peters et al. However, we can see the confusion in the paragraph so we have rewritten it as follows for clarity. “Scoping reviews provide a means of mapping key concepts in an area (Peters et al, 2015). Arksey and O’Malley first introduced a five-stage process to guide the conduct of scoping reviews in 2005. Levac, Colquhoun, and O’Brien advanced the scoping review methodology by providing clarification and enhancements for each of the five stages. They also introduced a sixth stage which suggested consultation with stakeholders for knowledge dissemination. Recently, in 2015, Peters et al, under the umbrella of the Joanna Briggs Institute (JBI), further clarified and standardized the methods associated with conducting a scoping review. Our review, then, was informed by Peters et al’s guidance on conducting scoping reviews.”

We developed an a priori protocol that guided our research, and it is available from the first author. We used the JBI mnemonic PCC: Participants (active athletes), Concept (stroke), Context (sports) to inform our search and data charting.

  1. The protocol mentioned in line 70, is it published or registered somewhere (e.g., OSF)?

We thank you for this comment. We agree that we should have published the protocol in a resource such as the Open Science framework, but we were new to the idea and had not had much exposure to OSF at the time. We will most certainly publish our protocol in OSF or other similar repository for the next scoping review we conduct a priori.

  1. Inclusion criteria are well described. But there are no exclusion criteria.

We included the following to clarify the exclusion criteria. “Studies that focused on non-athlete populations, or non-active athletes (e.g. retired) were excluded. Further, studies using sports as a form of rehabilitation were not included. Studies that did not include stroke, as defined above, were also excluded.  All reviews including systematic, scoping, and narrative were excluded.” We did exclude “the term “stroke” (line 111-112) since it is often used in sports literature, and we mindfully excluded sports-related stroke terms in the search, for example, golf stroke and swimming stroke. We also excluded medical terms such as heat stroke and stroke volume.”

  1. Please, avoid common expressions as 'aka' or 'etc', as they have no scientific meaning or soundness and shouldn't be used.

Thank you for your due diligence and based on The Cambridge Dictionary ‘aka refers to ‘also as known as’ and we have substituted this on line 92. However, etc., is a Latin expression used in English to mean "and other similar things", therefore we have not changed this as we believe it is both appropriate and a well-known/recognized convention.

  1. On '2.1 Definitions', authors state that they consider the definition of the stroke to include 'the simplest level when blood flow to the brain is hindered'. What about 'syncopes'? As you know, TIA can be incorrectly diagnosed as syncope’s, of which the main clinical feature is precisely the decrease in systemic arterial pressure and in the global cerebral blood flow. This is a huge source of bias.

Yes, we are aware that TIA, particularly postural or situational related to sport could result in an incorrect diagnosis of syncope. However, we reviewed each case and the diagnosis associated with it.  We have also added a statement related to this in the limitations. TIA is a known frequent explanation for syncope; however, these patients tend to be elderly males with a high incidence of ischemic heart disease and hypertension (Davidison et al., (1991). (Clin Cardiol 14, 141-144).

RESULTS

  1. Authors should explain the reason to exclude 4083 reports from their review.

We included the following to clarify (line 158)…  (4083 of which did not meet the inclusion criteria).

As noted in the PRISMA Flow Diagram 2020, there is no requirement to list the reasons for records excluded at the title/abstract screening phase. Additionally, Covidence does not provide a way to capture the reason for exclusion at title/abstract phase. A reason for exclusion at full-text phase, though, is required in the PRISMA Flow Diagram 2020, and is captured by Covidence.

  1. What do you mean by '39 cases meeting the inclusion criteria', if only 36 were included in the review?

Our apologies if this was not explicit enough. The article by McCrory (2000) reports on 3 cases bringing the total to 39 (36 articles +3 = 39 cases). Table A identifies this (no. 10 in the table, McCrory) with the reporting of 2 cases of Aussie Football and 1 case of Rugby League. 

  1. The description of the inciting event in Table 2 is difficult to read, as lines are too close to each other. Would it be possible to draw or format the Table for a better reading?

Unfortunately, the journal format of portrait versus landscape does result in a small font for reading. The reporting of the data extraction approach in Tables A and B follows the Joanna Briggs Institute (JBI) methodology ( a lot of information) and this includes in order: author, year of publication, and country of origin followed by study population (sport and age) and the inciting event. 

  1. The 7th report (Kersey, 1998): I suggest writing 'Persisted for 12 days', instead of 'persisted for 12 days' and avoid starting sentences with numbers ('5 weeks later').

Thank you for this correction. Now changed to “Persisted for 12 days and 5 weeks later…”

  1. The 8th report (Malek, 2000), the original authors report the kick boxer to have 42 years. Maybe you could include that.

Thank you for this suggestion. We have included his age and country of origin, both, overcites on our part.

DISCUSSION

  1. In the first paragraph the authors write an objective that's in accordance with the one presented in the Introduction, but not with the abstract.

Thank you we have modified the introduction to the discussion to be in line with our overarching goal and to assist the reader with the flow of the discussion. “This scoping review aimed to contribute and provide guidance to the ongoing development of a context-fit stroke model by summarizing the studies on a broad research topic related to stroke in sport based on a strict athlete definition. Prevention for stroke is known to broadly encompass three areas. These areas include lifestyle or modifiable risk factors, genetic predispositions or nonmodifiable risk factors, and trauma prevention factors such as sport or stroke triggers which is a relatively new area of investigation according to Boehme et al, (2017) [55]. In this discussion, these three areas are used to categorize the cases which are further grouped based on commonalities of the cases. This is followed by a summary of the case results highlighting some of the inciting events prior to stroke in active athletes, more frequent sport triggers, and inaccuracies in reporting of stroke.”

  1. Line 255: a closing parenthesis is missing after 'cardiovascular events'.

Thank you for catching this error. ‘(i.e., patent foramen ovalem [54,33]; atrial septal defect, [53] contributed to adverse cardiovascular events).

  1. Line 301: a typo in 'cerebellar region with can predispose'.

Thank you for pointing out this error. Changed to “the cerebellar region which…”

SUMMARY

  1. Line 381: a typo in 'hyper-rotation of the neck the (22%), leading'.

Thank you for identifying the extra ‘the’, which was deleted.

CONCLUSION

  1. This section should be revised, taking into consideration the possible changes suggested before, namely in the objectives. 

Thank you. We have considered your suggestions and have made changes related to the goals (aim) of the review and subsequently the discussion introduction which we believe improves the transparency of our approach.

  1. There should be something related to solutions/answers to the type of stroke and relation to sports.

We did make suggestions and highlighted these at the end of each category area in the discussion. However, see below.

  1. If EBP is to be maintained, some conclusions on this should also be written.

We have modified our wording related to this somewhat and briefly. Given the purpose of a scoping review is to provide an overview of the available research evidence without producing a summary answer to discrete research questions we believe our conclusion should remain the same. (Sucharew H, Maurizio Macaluso. Methods for Research Evidence Synthesis: The Scoping Review Approach. J Hosp. Med  2019, 7, 416-418. doi:10.12788/jhm.3248.

Reviewer 2 Report

Comments and Suggestions for Authors:

  • The authors must explain the methodology used in the abstract section
  • Authors are advised to use the word "stroke" instead of cerebrovascular accident throughout the entire article to avoid confusion
  • The protocol of the review has been previously registered. If not, justify the reason for not having done so
  • The bibliographic references throughout the text must be adapted to the recommendations of the journal
  • What is the reason that only articles in English are included? This fact may be an important limitation of this research
  • What is the meaning of the KAH? And JBI?
  • What in the meaning og (p.22) in line 69?
  • What is the search strategy used in this review? It should be indicated in the text and not as supplementary material. Supplementary material is not available
  • Was the search strategy used initially adapted to each of the databases? As written, it is not clear
  • What was the procedure for resolving discrepancies between reviewers? What was done if it was not achieved?
  • What is the reason why the calibration exercise was carried out by two people while the screening process was carried out by only one person? Both processes should have been carried out by two people to give greater consistency to the review?
  • Was the level of agreement between the two people who did the calibration exercise, the screening process and the selection of the articles measured? If not, the authors must justify the reason for not having done it.
  • The quality of the studies was taken into account for their selection. If yes, specify with which tool it was valued. If not, explain the reason for not having used it
  • The flow chart as it is constructed is very difficult to understand. Furthermore, it does not conform to the recommendations of the PRISMA statement. I advise the authors to redo it, taking into account the recommendations that they will find on the following web page: http://prisma-statement.org/prismastatement/flowdiagram.aspx
  • It is recommended that in the first part of the flow diagram the results obtained in each of the databases are specified. What is the reason why duplicate articles appear in the fourth phase if they had already been eliminated in the second phase?
  • The results section should be redone, specifying the main characteristics of the included studies as well as their results.
  • The discussion should be based on the results obtained in the study. It should not contain bibliographic citations. It is recommended to do it again
  • The discussion section hardly discusses the results obtained in the review. It should be redone with the objective of the study in mind. The sumary section should be removed and included in the first part of the discussion. The paragraph of genetic risk factors is difficult to understand; Authors should appreciate rewriting it
  • This review has important limitations that should be included in the discussion section. Current limitations should be extended
  • The conclusion should be based on the results obtained in the review, so it should be redone. It is advisable that there are no bibliographic references in this paragraph

Author Response

  • Thank you for your thorough review and suggestions to increase the readability of this manuscript, we recognize and appreciative the time a review takes.

The authors must explain the methodology used in the abstract section

  • The purpose of a scoping review is to provide an overview of the available research evidence without producing a summary answer to discrete research questions so we have modified our abstract and introduction/purpose. Sucharew H, Maurizio Macaluso, MD, DrPH, Methods for Research Evidence Synthesis: The Scoping Review Approach. J. Hosp. Med 2019;7;416-418. Published online first June 12, 2019.. doi:10.12788/jhm.3248

Authors are advised to use the word "stroke" instead of cerebrovascular accident throughout the entire article to avoid confusion

  • Thank you we have made these changes for the most part.

The protocol of the review has been previously registered. If not, justify the reason for not having done so

  • We thank the reviewer for their comment. We agree that we should have published the protocol in a resource such as the Open Science framework, but we were new to the idea and had not had much exposure to OSF at the time. We will most certainly publish our protocol in OSF or other similar repository for the next scoping review we conduct a priori.”

The bibliographic references throughout the text must be adapted to the recommendations of the journal

  • These have been completed based on the journal style and as far as we know have been checked by the editor.

What is the reason that only articles in English are included? This fact may be an important limitation of this research.

  • For feasibility reasons, our review was limited to English language studies only. The research team did not speak other languages so would be unable to translate the studies. Further, as the review was conducted during COVID, academic libraries around the world were closed, so it was difficult to obtain studies published in different languages. This is not an excuse just a pragmatic perspective. Please note that recent research has indicated that “Restricting systematic reviews to English-language publications appears to have little impact on the effect estimates and conclusions of systematic reviews.” (Dobrescu et al 2021); however, we have NOW indicated that the English language only studies is a limitation.
  • “Further, we included English language studies only, and could have potentially missed relevant studies published in other languages.”
  • Dobrescu, A. I., Nussbaumer, S. B., Klerings, I., Wagner, G., Persad, E., Sommer, I., ... & Gartlehner, G. (2021). Restricting evidence syntheses of interventions to English-language publications is a viable methodological shortcut for most medical topics: a systematic review: Excluding English-language publications a valid shortcut. Journal of Clinical Epidemiology.

What is the meaning of the KAH?

  • KAH are the initials of the health sciences librarian (KAH).
  • JBI refers to Joanna Briggs Institute (JBI) and we have added this in. 

What in the meaning og (p.22) in line 69?

  • We are not sure what this is referring to and can’t not find the occurrence “og”.

What is the search strategy used in this review? It should be indicated in the text and not as supplementary material. Supplementary material is not available.

  • The new PRISMA 2020 checklist and reporting guidelines, as well as the PRISMA-S (PRISMA Extension for searching), require the complete search strategies for all databases: Checklist states: Present the full search strategies for all databases, registers, and websites, including any filters and limits used. It would be beyond the word limits to include the search strategy for each database within the manuscript itself as the journal does not permit online supplements (apologies, we were not aware of this), we have added the Medline search strategy as an Appendix, at the end of the manuscript.  This is the format used by another recent scoping review published in this journal (Hasan, F., Marsia, S., Patel, K., Agrawal, P., & Razzak, J. A. (2021). Effective Community-Based Interventions for the Prevention and Management of Heat-Related Illnesses: A Scoping Review. International Journal of Environmental Research and Public Health, 18(16), 8362.)
  • A brief overview of the search strategy is provided in the manuscript, section 2.2 – 2.3; however, for transparency and reproducibility, it is not sufficient to list some keywords and subject headings.

Was the search strategy used initially adapted to each of the databases? As written, it is not clear

  • We have clarified the sentence: The search was first developed in Medline, and then was translated and adapted for each database. Keywords were the same across all databases, and subject headings were determined by the database indexing.
  •  
  • What was the procedure for resolving discrepancies between reviewers? What was done if it was not achieved?
  •  
  • Our protocol indicated that “A third expert will be included if consensus cannot be reached.”  However as consensus was reached on all discrepancies, a third expert was not required so we did not report on this in the manuscript.

What is the reason why the calibration exercise was carried out by two people while the screening process was carried out by only one person? Both processes should have been carried out by two people to give greater consistency to the review?

  • The PRISMA 2020 explanation and elaboration document (Page et al  (2021). PRISMA 2020 explanation and elaboration: updated guidance and exemplars for reporting systematic reviews. BMJ, 372.) notes that selecting of studies may be completed by one reviewer: “Single screening is an efficient use of time and resources but there is a higher risk of missing relevant studies” (p. 9, Box 3). Page et al cite three references to support single screening.  Further, in our manuscript, we cite The Cochrane Handbook for Systematic Reviews where it is noted that it is acceptable for one person to screen titles/abstracts.  Full-text screening was carried out by two researchers, again, as per the guidance from the Cochrane Handbook.
  • In order to ensure that the inclusion/exclusion criteria were clear and well defined, prior to starting title and abstract review, both screeners undertook a calibration exercise with a sample of 500 studies (approximately 10% of the total retrieved). JBI Scoping Review methodology recommends a random sample of titles and abstracts be screened as a pilot  “to discuss discrepancies and make modifications to the eligibility criteria and definitions/elaboration document” (https://wiki.jbi.global/display/MANUAL/11.2.6+Source+of+evidence+selection). As the same selection criteria are used for both title/abstract and full-text screening, it was imperative that both screeners participate in this calibration exercise where the aim was to ensure clear and precise inclusion/exclusion criteria, and that both screeners applied the selection criteria in the same way.

Was the level of agreement between the two people who did the calibration exercise, the screening process and the selection of the articles measured? If not, the authors must justify the reason for not having done it.

  • The proportionate agreement was 0.96 with a yes probability of .00185 and no probability of 0.91551. Random Agreement probability was 0.917. Cohen’s Kappa for full-text screening was .5869.

The quality of the studies was taken into account for their selection. If yes, specify with which tool it was valued. If not, explain the reason for not having used it

  • Critical appraisal is not a requirement of a scoping review, nor is it generally recommended. “Critical appraisal or risk of bias assessment is generally not recommended in scoping reviews because the aim is to map the available evidence rather than provide a synthesized and clinically meaningful answer to a question. For this reason, an assessment of methodological limitations or risk of bias of the evidence included within a scoping review is generally not performed (unless there is a specific requirement due to the nature of the scoping review aim).” (Peters et al, p. 2124). Peters, M. D., Marnie, C., Tricco, A. C., Pollock, D., Munn, Z., Alexander, L., ... & Khalil, H. (2020). Updated methodological guidance for the conduct of scoping reviews. JBI evidence synthesis18(10), 2119-2126.

The flow chart as it is constructed is very difficult to understand. Furthermore, it does not conform to the recommendations of the PRISMA statement. I advise the authors to redo it, taking into account the recommendations that they will find on the following web page: http://prisma-statement.org/prismastatement/flowdiagram.aspx

  • The PRISMA Flow diagram is the new PRISMA 2020 Flow Diagram, and the template was downloaded from here: http://prisma-statement.org/prismastatement/flowdiagram.aspx We used the PRISMA Flow Diagram 2020 for new systematic reviews (the first option from the link above) which includes searches of databases and registers only; however, we did adapt it slightly. We have now included a new version of our PRISMA Flow Diagram (2020) with the addition of the Records removed before screening box included, and have also now included the number of results per database (point below)

It is recommended that in the first part of the flow diagram the results obtained in each of the databases are specified. What is the reason why duplicate articles appear in the fourth phase if they had already been eliminated in the second phase?

  • The number of records retrieved from each database is now included in the PRISMA Flow Diagram. Covidence was used for the deduplication process.  “To identify duplicate references, Covidence checks on title, year, and volume, all of which must match exactly, and the authors must be similar.” (https://support.covidence.org/help/viewing-duplicates).  Given that the duplicates must match exactly on title, year and volume, there is the possibility of some duplicates remaining. Often these duplicates are not discovered until full text screening, and sometimes until data extraction.
  • Covidence is a sound choice for automated deduplication. A recent study indicated that Covidence demonstrated the highest specificity for identifying duplicates and that “Another considerable strength of de-duplicating references using Ovid multifile search or Covidence is that the process is fully automated, so duplicate references are automatically removed from the unique references and user mediation was not necessitated.” (p. 6). McKeown, S., & Mir, Z. M. (2021). Considerations for conducting systematic reviews: evaluating the performance of different methods for de-duplicating references. Systematic Reviews10(1), 1-8.

The results section should be redone, specifying the main characteristics of the included studies as well as their results.

  • We appreciate your perspective but believe we have discussed the main characteristics of the included studies

The discussion should be based on the results obtained in the study. It should not contain bibliographic citations. It is recommended to do it again

  • The discussion outlines the results of the study (therefore must include the references of these studies at a minimum and the supporting studies) and the discussion flow used the headings (see below the introductory statement of the discussion) to discuss the cases reported on. “Prevention for stroke broadly encompasses the areas of lifestyle or modifiable risk factors, genetic predispositions or nonmodifiable risk factors, and trauma prevention factors such as sport or stroke triggers which is a relatively new area of investigation according to Boehme et al, (2017) [55].”

The discussion section hardly discusses the results obtained in the review. It should be redone with the objective of the study in mind. The paragraph of genetic risk factors is difficult to understand;

  • We appreciate your perspective which seems somewhat contradictory to your above comment.

The summary section should be removed and included in the first part of the discussion.

  • Some discussions may begin with a summary however we have chosen to have a summary at the end of our discussion.

This review has important limitations that should be included in the discussion section. Current limitations should be extended.

  • Thank you, we have added a few other bias’ including English only as a limitation.

The conclusion should be based on the results obtained in the review, so it should be redone. It is advisable that there are no bibliographic references in this paragraph

  • Respectively, conclusions often do not have citations.

Reviewer 3 Report

The article entitled “Stroke and Athletes: A Scoping Review” reviewed nicely the incidence of stroke in athletes which is an interesting topic for being less discussed. Additionally, few cases on this topic are available. The major content of this review is based on case reports and to a lesser extent on a research article. It is very well organized and discussed a variety of stroke triggers in athletes including sport-related traumatic events and other predisposing factors including genetic factors, supplements, stimulants, anabolic steroids... etc. I have some minor issues that need to be considered:       

  1. The tables need rearrangement as it is not clear to which point the inciting events are belonging. I don’t think it is necessary to mention the date, author, and country of the case report; instead it is better to classify cases depending on the type of sport and explain the inciting events for each group.
  2. In table B, dots should be placed at the end of all sentences.
  3. In line 301, “with” in (the cerebellar region with can predispose) should be replaced with “which”. In line 381, “the” in (of the neck the) should be removed.

Author Response

  • Thank you for your review and suggestions to increase the readability of this manuscript.
  1. The tables need rearrangement as it is not clear to which point the inciting events are belonging. I don’t think it is necessary to mention the date, author, and country of the case report; instead it is better to classify cases depending on the type of sport and explain the inciting events for each group.
  • Thank you for your suggestions regarding the information within the tables. The reporting of the data extraction approach in Tables A and B follows the Joanna Briggs Institute (JBI) methodology and this includes in order: author, year of publication, and country of origin followed by study population (sport and age) and the inciting event. The type of study design (case report and newspaper/magazine/article) is embedded in the table title. We note that case studies particularly older ones may have issues of reporting both incomplete and not necessarily transparent when it comes to the inciting event. Therefore we gave a description to point in the direction of the inciting event. We did not want to specifically label inciting events because we would be making some assumptions. There may be other reasons for the cases to not identify inciting events. For example, delayed symptomology is underreported as we stated earlier in the manuscript. “Athletes not reporting symptoms is a reoccurring theme in the literature that has been particularly noted in high-risk contact sports such as football where an image of toughness has been part of the culture. Recently, education and awareness of concussion etiology and symptoms or initial lack of it has been improving in the younger age groups [94]. Education, however on headache symptomology such as a precipitating headache or migraine associated with CVA may be lacking in athletes based on this review.”
  • We also recognize that the journal publication format (portrait versus landscape) does make the font size small.

  1. In table B, dots should be placed at the end of all sentences.
  • Thank you for identifying the inconsistencies in the periods at the end of the sentence’s typos; changed.
  1. In line 301, “with” in (the cerebellar region with can predispose) should be replaced with “which”. In line 381, “the” in (of the neck the) should be removed.
  • Thank you for identifying these typos; changed.

Round 2

Reviewer 1 Report

Desta authors,

Congratulations on your work revising the manuscript. I hope my sugestions and comments were useful. The manuscript is greatly improved.

Author Response

Thank you so much for your suggestions and comments. 

Reviewer 2 Report

Comments and suggestions for authors

  • The authors have modified the purpose of the study in the abstract, but what the reviewer requested was that they include a better explanation of the methodology used.
  • The reviewer asked if the study protocol had been previously registered, such as in PROSPERO. The reviewer did not ask whether the protocol had been published or not. Authors should explain why the review protocol has not been published
  • Restricting the search only to the English language significantly reduces the generalization of the results and conclusions obtained.
  • The reviewer encourages authors to include the full search strategy used in each database in table form.
  • The results section should be redone, specifying the main characteristics of the included studies, as well as their result
  • The discussion should be based on the results obtained in the study, so it is advisable to review it

Author Response

Thank you for recognizing the changes and modifications we make. We have made a few more but also provide more explanation for why we did not make other changes.

  • The authors have modified the purpose of the study in the abstract, but what the reviewer requested was that they include a better explanation of the methodology used.

See below.

  • The reviewer asked if the study protocol had been previously registered, such as in PROSPERO.

I believe we were confused by this statement since scoping review protocols are not accepted by Prospero.

  • The reviewer did not ask whether the protocol had been published or not. Authors should explain why the review protocol has not been published.

We recognize now that it is best practice to submit one’s scoping review protocol to a registry, such as Open Science Framework or an institutional repository. We were mistaken in not submitting our a priori scoping review protocol to a registry. However, the protocol was written prior to the research being undertaken and is available from the first author. If the reviewer suggests, we can submit the a priori protocol to our Institutional Repository now. Unfortunately, we cannot go back in time to rectify our oversight.

Therefore we have a made change to read on L72 “The protocol was developed a priori to guide the scoping review; however, it was not registered. It is available from the first author.”

We also conducted a quick analysis of other current scoping reviews in this journal International Journal of Environmental Research and Public Health. We reviewed the first page of retrieved results (50 studies) that were found with the search: scoping in the title.

It appears that it, for the International Journal of Environmental Research and Public Health is actually uncommon for a protocol to be written, let alone published open access a priori for scoping reviews in this journal. 

Ruppert R, Kattari SK, Sussman S. Review: Prevalence of Addictions among Transgender and Gender Diverse Subgroups. International Journal of Environmental Research and Public Health. 2021; 18(16):8843. https://doi.org/10.3390/ijerph18168843

  • No mention of a protocol written prior to undertaking the scoping review.

Lange S, MÄ™drzycka-DÄ…browska W, Friganovic A, Oomen B, Krupa S. Delirium in Critical Illness Patients and the Potential Role of Thiamine Therapy in Prevention and Treatment: Findings from a Scoping Review with Implications for Evidence-Based Practice. International Journal of Environmental Research and Public Health. 2021; 18(16):8809. https://doi.org/10.3390/ijerph18168809

  • No mention of a protocol written prior to undertaking the scoping review.

Toro-Alzate L, Hofstraat K, de Vries DH. The Pandemic beyond the Pandemic: A Scoping Review on the Social Relationships between COVID-19 and Antimicrobial Resistance. International Journal of Environmental Research and Public Health. 2021; 18(16):8766. https://doi.org/10.3390/ijerph18168766

  • This study notes: “Because it was seen as a scoping review, registration of the review protocol was not necessary.” (p. 3/20)

Harada T, Miyagami T, Kunitomo K, Shimizu T. Clinical Decision Support Systems for Diagnosis in Primary Care: A Scoping Review. International Journal of Environmental Research and Public Health. 2021; 18(16):8435. https://doi.org/10.3390/ijerph18168435

  • No mention of a protocol written prior to undertaking the scoping review.

Hasan F, Marsia S, Patel K, Agrawal P, Razzak JA. Effective Community-Based Interventions for the Prevention and Management of Heat-Related Illnesses: A Scoping Review. International Journal of Environmental Research and Public Health. 2021; 18(16):8362. https://doi.org/10.3390/ijerph18168362

  • No mention of a protocol written prior to undertaking the scoping review.

Filho VCB, Pereira WMG, Farias BdO, Moreira TMM, Guerra PH, Queiroz ACM, Castro VHSd, Silva KS. Scoping Review on Interventions for Physical Activity and Physical Literacy Components in Brazilian School-Aged Children and Adolescents. International Journal of Environmental Research and Public Health. 2021; 18(16):8349. https://doi.org/10.3390/ijerph18168349

  • 1. Protocol

This scoping review followed the recommendations of the Joanna Briggs Institute manual [13] and was reported in accordance with the guidelines recommended by the Preferred Reporting Items for Systematic Review and Meta-Analyses Extension for Scoping Reviews (PRISMA-ScR) [14], as detailed in Table S1. However, this study was not registered. (p. 2/16)

Moebus S, Boedeker W. Case Fatality as an Indicator for the Human Toxicity of Pesticides—A Systematic Scoping Review on the Availability and Variability of Severity Indicators of Pesticide Poisoning. International Journal of Environmental Research and Public Health. 2021; 18(16):8307. https://doi.org/10.3390/ijerph18168307

  • We conducted a systematic literature review without prior protocol by starting the search for publications in the database PUBMED.(p. 3/15)

Rashidi A, Whitehead L, Newson L, Astin F, Gill P, Lane DA, Lip GYH, Neubeck L, Ski CF, Thompson DR, Walthall H, Jones ID. The Role of Acceptance and Commitment Therapy in Cardiovascular and Diabetes Healthcare: A Scoping Review. International Journal of Environmental Research and Public Health. 2021; 18(15):8126. https://doi.org/10.3390/ijerph18158126

  • No mention of a protocol written prior to undertaking the scoping review.

Kuipers S, Boonstra N, Kronenberg L, Keuning-Plantinga A, Castelein S. Oral Health Interventions in Patients with a Mental Health Disorder: A Scoping Review with Critical Appraisal of the Literature. International Journal of Environmental Research and Public Health. 2021; 18(15):8113. https://doi.org/10.3390/ijerph18158113

  • This study did not require ethical approval, as data was collected from existing published peer-reviewed literature and grey literature. The protocol for this review was registered in PROSPERO (ID: CRD42018114415).

NOTE: Prospero does not accept scoping review protocols; within the document, the authors refer to their study as a “systematic scoping review”.

Michael V, You YX, Shahar S, Manaf ZA, Haron H, Shahrir SN, Majid HA, Chia YC, Brown MK, He FJ, MacGregor GA. Barriers, Enablers, and Perceptions on Dietary Salt Reduction in the Out-of-Home Sectors: A Scoping Review. International Journal of Environmental Research and Public Health. 2021; 18(15):8099. https://doi.org/10.3390/ijerph18158099

  • No mention of a protocol written prior to undertaking the scoping review.

Cornelius SL, Berry T, Goodrich AJ, Shiner B, Riblet NB. The Effect of Meteorological, Pollution, and Geographic Exposures on Death by Suicide: A Scoping Review. International Journal of Environmental Research and Public Health. 2021; 18(15):7809. https://doi.org/10.3390/ijerph18157809

  • No mention of a protocol written prior to undertaking the scoping review.

Mau M, Aaby A, Klausen SH, Roessler KK. Are Long-Distance Walks Therapeutic? A Systematic Scoping Review of the Conceptualization of Long-Distance Walking and Its Relation to Mental Health. International Journal of Environmental Research and Public Health. 2021; 18(15):7741. https://doi.org/10.3390/ijerph18157741

  • The protocol was registered in the Open Science Framework in February 2021 and is available online through https://osf.io/cnhys/(accessed on 16 July 2021).
  • Restricting the search only to the English language significantly reduces the generalization of the results and conclusions obtained.

We previously provided a detailed explanation, a supporting reference, and an additional statement (L436-467) in the manuscript to the comment regarding the reviewer’s question on the reason that only articles in English were included and that this respectfully maybe (it is not a fact) an important limitation of this research.

We can further support that by adding that excluding non-English publications from evidence-syntheses is likely to not change conclusions according to a recent study by Nussbaumer-Streit et al., (2020 (Journal of Clinical Epidemiology V 118 DOI: 10.1016/j.jclinepi.2019.10.011). In their study result, 38 of the 40 outcomes, [non-English studies excluded] did not markedly alter the size or direction of effect estimates or statistical significance. 

  • The reviewer encourages authors to include the full search strategy used in each database in table form.

We have included the full search as a supplement. Previously stated we would be over the word limit if we considered this. We note it would be very unusual to find the full search strategy used in each database as a table in text.

  • The results section should be redone, specifying the main characteristics of the included studies, as well as their results
  • The discussion should be based on the results obtained in the study, so it is advisable to review it.

Respectfully we have addressed this. Perhaps the reviewer would clarify beyond what we completed as the main characteristics given we have mapped out the 'breath, nature and extent of research'. This includes Population-athlete, Concept-stroke, and Context-sport.

Finally, both other reviewers have signed off on the changes and we wonder if their credibility is called into question given that one reviewer has stated that this manuscript requires extensive editing of English language and style.  No examples were identified.